# Biophysicochemical motifs in T cell receptor sequences as a potential biomarker for high-grade serous ovarian carcinoma

**Jared Ostmeyer**[1], **Elena Lucas**[2], **Scott Christley**[3], **Jayanthi Lea**[4], **Nancy Monson**[5], **Jasmin Tiro**[6], **Lindsay G. Cowell**[7]*

**1** Department of Population and Data Sciences, UT Southwestern Medical Center, Dallas, TX, United States of America, **2** Department of Pathology, UT Southwestern Medical Center, Dallas, TX, United States of America, **3** Department of Population and Data Sciences, UT Southwestern Medical Center, Dallas, TX, United States of America, **4** Department of Obstetrics and Gynecology, UT Southwestern Medical Center, Dallas, TX, United States of America, **5** Department of Neurology and Neurotherapeutics, Department of Immunology, UT Southwestern Medical Center, Dallas, TX, United States of America, **6** Department of Population and Data Sciences, UT Southwestern Medical Center, Dallas, TX, United States of America, **7** Department of Population and Data Sciences, Department of Immunology, UT Southwestern Medical Center, Dallas, TX, United States of America

* lindsay.cowell@utsouthwestern.edu

**Data Availability Statement:** All sequence data are freely available from the VDJServer Community Data Portal (CDP) (vdjserver.org) under the project

## Abstract

We previously showed, in a pilot study with publicly available data, that T cell receptor (TCR) repertoires from tumor infiltrating lymphocytes (TILs) could be distinguished from adjacent healthy tissue repertoires by the presence of TCRs bearing specific, biophysico-chemical motifs in their antigen binding regions. We hypothesized that such motifs might allow development of a novel approach to cancer detection. The motifs were cancer specific and achieved high classification accuracy: we found distinct motifs for breast versus colorectal cancer-associated repertoires, and the colorectal cancer motif achieved 93% accuracy, while the breast cancer motif achieved 94% accuracy. In the current study, we sought to determine whether such motifs exist for ovarian cancer, a cancer type for which detection methods are urgently needed. We made two significant advances over the prior work. First, the prior study used patient-matched TILs and healthy repertoires, collecting healthy tissue adjacent to the tumors. The current study collected TILs from patients with high-grade serous ovarian carcinoma (HGSOC) and healthy ovary repertoires from cancer-free women undergoing hysterectomy/salpingo-oophorectomy for benign disease. Thus, the classification task is distinguishing women with cancer from women without cancer. Second, in the prior study, classification accuracy was measured by patient-hold-out cross-validation on the training data. In the current study, classification accuracy was additionally assessed on an independent cohort not used during model development to establish the generalizability of the motif to unseen data. Classification accuracy was 95% by patient-hold-out cross-validation on the training set and 80% when the model was applied to the blinded test set. The results on the blinded test set demonstrate a biophysicochemical TCR motif found overwhelmingly in women with HGSOC but rarely in women with healthy ovaries, strengthening the proposal that cancer detection approaches might benefit from incorporation of TCR

accession 3276777473314001386-242ac116-0001-012.

**Funding:** This project was supported by funding to LGC from UT Southwestern Medical Center, Be the Difference Foundation, Commercial Real Estate Women of Dallas (CREW Dallas), and an anonymous donor. CREW Dallas is NOT a commercial entity. It is a 501c3. The funders had no role in study design, data collection and analysis, decision to publish, or preparation of the manuscript.

**Competing interests:** NO authors have competing interests. The authors are not aware of any competing interests.

motif-based biomarkers. Furthermore, these results call for studies on large cohorts to establish higher classification accuracies, as well as for studies in other cancer types.

## Introduction

Despite the tremendous genomic heterogeneity between cancers, there is evidence that cancer patients mount T cell responses against antigens they have in common, including tumor antigens. Shared tumor antigens can be generally classified into three categories: (1) self-antigens with dysregulated expression or increased copy numbers, such as MelanA, HER2, SOX2, and NY-ESO-1 [1–5], (2) altered self-antigens, such as recurrent oncogenic mutations, including $BRAF_{V600E}$ and $CDK_{R24C}$ [6] and TGF-βRII frameshift mutations [7], and (3) non-self-antigens–viral epitopes expressed by virus-induced cancers, such as those derived from Human Papilloma Virus [8, 9], Hepatitis B Virus [10], and Epstein Barr Virus [11]. Ovarian cancer is considered rich in the first category of shared tumor antigens, with relatively large percentages of ovarian cancers expressing MAGE-A1, MAGE-A3, NY-ESO-1, and others [12, 13]. In the case of the alpha folate receptor, 97% of ovarian cancers were found to express it, with the vast majority having moderate or strong expression levels, while only 63% of healthy ovaries were found to express it, and in all cases the expression was weak [14].

Evidence for T cell responses against shared tumor antigens comes from studies demonstrating the presence of T cells with binding capacity for, and reactivity to, the shared antigens [1, 3, 4, 15–20]. Indeed, responses against shared tumor antigens may outnumber those against mutated neoantigens, including for highly mutated cancers such as melanoma [21, 22]. In addition to effector T cells responding to tumor antigens, a significant portion of the tumor-infiltrating lymphocyte (TIL) population is expected to be regulatory T cells that are reactive to tissue-restricted self-antigens associated with the organ of cancer origin, as these T cells are highly enriched in cancer lesions [21]. Thus, on balance, we expect much of a TIL population to be composed of T cells with specificity for antigens shared across cancer patients and not present, or present at significantly reduced levels, in cancer-free individuals.

We hypothesized that the above-described T cell responses could serve as the basis for cancer early detection biomarkers and sought to develop a method for detecting them that didn't require knowledge of the target antigens and didn't rely on the assumption that T cells responding to a common target would express T cell receptors with the same amino acid sequence. Utilizing publicly available TCR deep sequencing data, we applied multiple instance learning (MIL) after converting the TCR amino acid sequences to a biophysicochemical representation using Atchley Factors [23–25]. We found that TCR repertoires from breast or colorectal cancer TILs could be distinguished from adjacent healthy tissue repertoires by the presence of TCRs bearing specific, biophysicochemical motifs in their antigen binding regions [25]. The motifs were different between the two cancer types, and both achieved high classification accuracy. The colorectal cancer motif achieved 93% accuracy, while the breast cancer motif achieved 94% accuracy.

In the current study, we sought to establish the plausibility of using TCR motifs for ovarian cancer detection and applied our method to locally collected patient samples. We made two significant advances over the prior work. First, the prior study used patient-matched TILs and healthy repertoires, collecting healthy tissue adjacent to the tumors. Thus, the classification task was to distinguish two repertoires that had both been collected from an organ effected by cancer, one repertoire from within the cancerous lesion and one repertoire from a lesion-free region. The current study collected TILs from patients with high-grade serous ovarian carcinoma (HGSOC) and collected healthy ovary repertoires from cancer-free women undergoing hysterectomy with salpingo-oophorectomy for benign disease. Thus, the current classification

task is distinguishing repertoires from women with HGSOC versus repertoires from women with healthy ovaries. The second advance comes from the opportunity to assess the motif on a blinded test data set. In the prior study, only a training data set was available, and classification accuracy was measured by patient-hold-out cross-validation. In the current study, both a training and test data set were available. Thus, in addition to assessing classification accuracy of the motif by patient-hold-out cross-validation, the ability of the motif to generalize to a new, independent cohort of data not used for motif discovery was assessed.

The current study revealed a TCR biophysicochemical motif present overwhelmingly in HGSOC TILs repertoires but rarely in healthy ovary repertoires. The motif is specific to HGSOC, i.e., it is different from the motifs previously identified for colorectal and breast cancer. The classification accuracy assessed by cross-validation on the training data was 95% (19/ 20). Applying the same model selection and cross-validation procedure to data with permutated labels resulted in an average classification accuracy of 55%, and the accuracies of all 20 permutations were < 95%. Application of the best model to the unseen test set resulted in a classification accuracy of 80% (16/20), indicating that the motif has some capacity to generalize. These results strengthen the proposal that cancer detection approaches might benefit from incorporation of TCR motif-based biomarkers and call for studies assessing the approach on large training and testing data sets and on additional cancer types.

## Materials and methods

### Datasets

We obtained 40 archived tissue blocks from the Pathology Laboratories of Parkland Health and Hospital System and two university hospitals associated with UT Southwestern Medical Center (St. Paul Hospital and Clements University Hospital): 20 HGSOC specimens and 20 normal ovary specimens. The study was approved by the UT Southwestern Medical Center IRB, study number STU-2018-0239, with a waiver of consent because anonymized, archived, FFPE tissue blocks were used. We divided the blocks into two cohorts, one training cohort (Cohort I) and one test cohort (Cohort II), each with 10 HGSOC specimens and 10 normal ovary specimens (Table 1). In both cohorts, all donors were between 50 and 59 years of age. In the training cohort, eight of the HGSOC samples were stage IIIC, and two were stage IVB. The test cohort was more heterogeneous with respect to the HGSOC stage. Six were stage IIIC; one was stage IVB. The remaining three samples were stages IIA, IIB, and IIIA1(i). Of the 20 control samples, in addition to normal ovarian tissue, 12 had fallopian tube tissue within the block from which our samples were cut. In seven of the remaining eight cases, ovarian sections demonstrated serous (tubal-type) epithelial inclusions. Thus, these controls are representative of the tissue from which ovarian cancer is believed to arise. Tissue curls were sent to Adaptive Biotechnologies for sequencing of the TCR β (TCRB) locus at survey depth. Sequencing was based on genomic DNA, and, for the tissue blocks with fallopian tube tissue or epithelial inclusions, the DNA from the various tissue types was combined for sequencing. We did not quantify the number of TILs present in the tissue by immunohistochemistry prior to sequencing, but the number of unique TCR in Table 1 estimates the number of T cell clones present the sequencing sample [26, 27]. The study design is shown in Fig 1A.

The data are freely available from the VDJServer Community Data Portal (CDP) (vdjserver. org) under the project accession 3276777473314001386-242ac116-0001-012 [28]. The sequences are available in FASTA format in the "Browse Project Data" section. Annotated alignments are available in the tab-separated-values format recommended by the Adaptive Immune Receptor Repertoire Community in the "View Analyses and Results" section as output from IgBlast [29, 30].

**Table 1. Patient characteristics.** Age, stage, patient diagnosis, and the number of unique TCRB sequences for each sample in the training and validation cohorts.

| | | Age | FIGO Stage | Diagnosis | Unique TCRBs |
|---|---|---|---|---|---|
| **Cohort I, Training Cohort** | **HGSOC Cases** | 52 | IVB | High-grade serous carcinoma | 8353 |
| | | 55 | IIIC | High-grade serous carcinoma | 1343 |
| | | 58 | IVB | High-grade serous carcinoma | 3249 |
| | | 52 | IIIC | High-grade serous carcinoma | 2692 |
| | | 50 | IIIC | High-grade serous carcinoma with endometrioid component | 719 |
| | | 53 | IIIC | High-grade serous carcinoma | 2225 |
| | | 53 | IIIC | High-grade serous carcinoma | 7363 |
| | | 55 | IIIC | High-grade serous carcinoma | 1667 |
| | | 59 | IIIC | High-grade serous carcinoma | 190 |
| | | 52 | IIIC | High-grade serous carcinoma | 227 |
| | **Normal Ovary Cases** | 52 | - | Cervix with LSIL | 695 |
| | | 51 | - | Cervix with LSIL; uterus with LM, AM | 603 |
| | | 55 | - | Uterus with LM, AM | 1870 |
| | | 55 | - | Uterus with LM, AM | 780 |
| | | 53 | - | Uterus with DPE, LM, AM | 3788 |
| | | 51 | - | Uterus with LM, AM; contralateral ovary with EM | 2896 |
| | | 58 | - | Contalateral ovary with MCT | 1101 |
| | | 55 | - | Uterus with LM, AM | 337 |
| | | 51 | - | Uterus with AM | 1409 |
| | | 52 | - | Uterus with LM, AM | 423 |
| **Cohort II, Test Cohort** | **HGSOC Cases** | 51 | IIIC | High-grade serous carcinoma | 467 |
| | | 56 | IVB | High-grade serous carcinoma | 1562 |
| | | 57 | IIIA1(i) | High-grade serous carcinoma | 572 |
| | | 57 | IIIC | High-grade serous carcinoma | 2414 |
| | | 51 | IIIC | High-grade serous carcinoma | 1134 |
| | | 56 | IIA | High-grade serous carcinoma | 1036 |
| | | 55 | IIIC | High-grade serous carcinoma | 2532 |
| | | 54 | IIB | High-grade serous carcinoma | 398 |
| | | 57 | IIIC | High-grade serous carcinoma | 2287 |
| | | 51 | IIIC | High-grade serous carcinoma | 332 |
| | **Normal Ovary Cases** | 51 | - | Uterus with LM | 803 |
| | | 50 | - | Uterus with LM, AM | 1290 |
| | | 50 | - | Uterus with LM | 1285 |
| | | 50 | - | Uterus with LM | 807 |
| | | 55 | - | Uterus with LM | 439 |
| | | 52 | - | Uterus with LM, AM | 685 |
| | | 53 | - | Uterus with LM | 1708 |
| | | 50 | - | Uterus with LM | 152 |
| | | 50 | - | Uterus with LM | 1405 |
| | | 57 | - | Uterus with LM | 202 |

LM: Leiomyoma; AM: Ademomyosis; DPE: disordered proliferative endometrium; MCT: mature cystic teratoma; EM: endometriosis; LSIL: low-grade squamous intraepithelial lesion.

## Representing TCRs

As previously described [25], we analyzed X–ray crystallographic structures of human TCRs bound to peptide–MHC complex obtained from the Protein Data Bank in order to determine how to represent TCRB sequence in a way that would capture the antigen binding capabilities

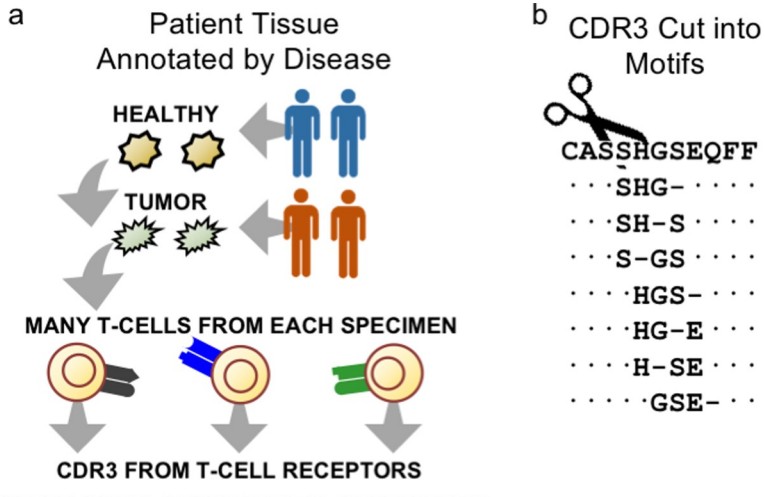

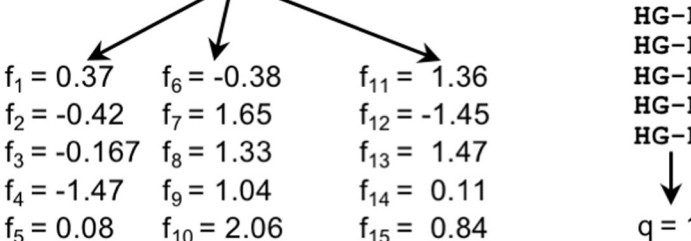

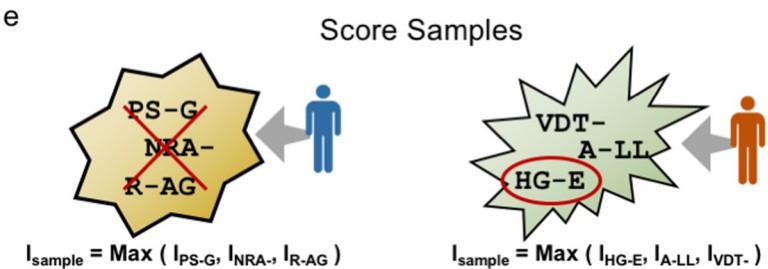

**Fig 1. Study overview.** (a) Ovarian samples are collected from patients with and without HGSOC cancer. High-throughput immune receptor sequencing reveals the TCRβ CDR3 sequences found in each tissue sample. (b) The CDR3 sequences are cut in motifs. In this example, a motif is assembled from three amino acid residues. Only a single residue from the CDR3 may be skipped, allowing for a single gap. Otherwise, the amino acid residues are contiguous neighbors. (c) Each amino acid residue is converted into a set of five chemical features using Atchley factors for a total of fifteen features describing the motif. The relative abundance of each motif is included as an additional sixteenth feature. (d) Each feature is multiplied by a weight ($\beta_1$ through $\beta_{16}$) that determines its relative importance, and a bias value ($\beta_0$) is added to calculate a logit. The logit can be converted into a probability value for that motif. (e) The weights and bias value are picked such that there is at least one motif with a probability value close to 1 in each HGSOC sample and all motifs in each healthy ovary sample have a probability close to 0.

of the corresponding TCRB chain. We focused on complementarity determining region 3 (CDR3), because it is the somatically generated portion of the gene and the primary determinant of the chain's antigen-binding specificity. We also focused on residues that directly contact peptide in a peptide–MHC complex. The crystal structure analysis revealed that TCRB CDR3 residues in contact with peptide tend to lie near each other, forming a local neighborhood of contact residues. The size and relative location of this neighborhood varied, but it rarely included any of the first or last three CDR3 residues, its average length was four, and in ~25% of cases, a non-contact residue was interspersed between the contact residues. Thus, to capture CDR3 contact residues, we excluded the first and last three CDR3 residues and partitioned the remaining sequence into every possible contiguous strip of three amino acid residues, referred to as a motif (Fig 1B). We also allowed one residue in the CDR3 sequence to be skipped when assembling a motif (Fig 1B). Such skipped residues are referred to as a gap. Our expectation is that, for each TCRB CDR3, at least one of its motifs contains residues that contact the peptide component of the receptor's cognate antigen. Alternative models were considered but exhibited reduced performance (Table 2).

When different TCRs bind the same peptide, the TCRB CDR3 contact residues may be different amino acids across the different TCRs. Thus, to identify motifs with different amino acid sequences but similar antigen-binding capabilities, we represented each motif using numerical values for the biophysicochemical properties of its component amino acids. We used Atchley factors as the biophysicochemical descriptors [24]. Atchley factors were derived from a set of over 50 amino acid properties by identifying clusters of properties that co-vary. The five Atchley factor values for each amino acid residue correspond loosely to its polarity, secondary structure, molecular volume, codon diversity, and electrostatic charge. For input into our model, each amino acid residue was represented by a vector of its five Atchley factor values (Fig 1C). With three amino acid residues in a motif, there are a total of 15 Atchley factor values that we represent using the symbols $f_1$ through $f_{15}$.

T-cells undergo clonal expansion in response to antigen stimulation, creating copies of the T-cell and its receptor. Thus, the quantity of a motif can indicate whether receptors containing it have encountered their cognate antigen. We therefore included an estimator of motif quantity as a feature in the model. When calculating the relative abundance of a motif in each sample, we identified every TCRB sequence containing the motif in its CDR3 and summed over the sequences' template counts, $C^{\text{CDR3}}$. This provided the motif count, $C^{\text{motif}}$. We then divided by the total count of all motifs in the sample, $T$, to get the motif's relative abundance, $f_q$.

$$C^{\text{motif}} = \sum_{\substack{\text{CDR3s} \\ \text{with motif}}} C^{\text{CDR3}} \;\; ; \;\; T = \sum_{\substack{\text{all motifs} \\ \text{from sample}}} C^{\text{motif}} \;\; ; \;\; f_q = \frac{C^{\text{motif}}}{T}$$

As with most statistical classifiers, it is important to normalize the input into the model, i.e., the model features. For each feature, we calculated a weighted mean and variance of the feature values over all motifs, where the weights were the relative abundances of each motif, $f_q$. Thus, motifs that appear more frequently exerted a greater influence on feature mean and variance than motifs that appeared only once or a few times. We then subtracted the mean from each feature value and divided by the square root of the variance to obtain a normalized value for model input.

## Logistic model

The 15 Atchley factor values for the three motif residues along with the motif's relative abundance were combined into a single vector $[f_1, f_2, \ldots, f_{15}, f_q]$ representing the features of a motif.

**Table 2. Different model configurations evaluated on Cohort 1.** Each row represents a different model, and the columns describe the configuration of each model. The first row (bold font) corresponds to the model configuration with the best performance for the breast and colorectal cancer datasets [25]. The second row (bold underlined font) corresponds to the best performing model configuration presented here. The first column indicates the number of amino acid residues in the motif. The second column indicates the number of CDR3 amino acid residues that could be skipped when assembling a motif. For example, if the value is 2, then 2 CDR3 amino acid residues could be skipped. The third column indicates if binary indicators indicating whether the corresponding CDR3 residue was ignored were used. For example, if a CDR3 residue was ignored but would have been in the third position of a motif if it had been included, then the 3rd indicator would have a value of 1. The fourth column indicates if an amino acid was skipped in the CDR3 for the given position in the motif. The fifth column indicates if the expected frequency of the motif in blood was included as a feature. The expected frequency was estimated using publicly available data from 786 presumed healthy individuals [31]. The sixth column indicates if the log of the motif relative abundance was used for the relative abundance term. Column 7 indicates if each feature is squared and used as an additional feature, resulting in 2nd order terms in the model. Column 8 indicates if batch normalization was used. Column 9 (fourth from last) is the log-loss averaged across the one-holdout cross-validations. Column 10 (third from last) is the accuracy computed over the one-holdout cross-validations. Column 11 (second from last) is the number of gradient steps used to fit the model as determined by early-stopping. Column 12 is the number of fits to the training data, of which the best fit to the training data is applied to the holdout sample.

| FEATURES | | | | | | | | CROSS-VAL LOG-LOSS | CROSS-VAL ACCUR-ACY | EARYL STOPP-ING | NUM FITS TO TRAIN |
|---|---|---|---|---|---|---|---|---|---|---|---|
| Motif Size | # of Gap Positions | One-Hot Indicator of Gap Position | Restricting Gap to Position X | Expected Frequency in Blood | Log Frequency Instead | 2nd Order Terms | Batch Norm. | | | | |
| **4** | **0** | | | | √ | | | **0.666** | **90%** | **1211** | **131072** |
| ***3*** | ***1*** | | | | | | | ***0.332*** | ***95%*** | ***2499*** | ***131072*** |
| 4 | 0 | | | √ | | | √ | 0.680 | 75% | 9 | 131072 |
| 3 | 1 | | | √ | | | | 0.400 | 95% | 1687 | 131072 |
| 4 | 0 | | | √ | | | | 0.887 | 65% | 1506 | 524288 |
| 3 | 1 | √ | | √ | | | | 0.477 | 90% | 1411 | 786432 |
| 3 | 2 | √ | | √ | | | | 0.963 | 65% | 467 | 131072 |
| 4 | 1 | √ | | √ | | | | 1.004 | 55% | 692 | 65536 |
| 4 | 2 | √ | | √ | | | | 0.639 | 80% | 3222 | 131072 |
| 3 | 0 | | | | | | | 1.083 | 50% | 3 | 131072 |
| 4 | 0 | | | | | | | 1.037 | 50% | 4 | 131072 |
| 3 | 1 | | x = 1 | | | | | 1.043 | 50% | 1 | 131072 |
| 3 | 1 | | x = 2 | | | | | 1.089 | 50% | 1 | 131072 |
| 3 | 1 | | x = 3 | | | | | 1.072 | 50% | 4 | 131072 |
| 3 | 1 | √ | | | | | | 0.378 | 90% | 2499 | 786432 |
| 3 | 2 | √ | | | | | | 1.016 | 75% | 1145 | 131072 |
| 4 | 0 | | | √ | | √ | | 1.083 | 50% | 5 | 131072 |
| 3 | 1 | √ | | √ | | √ | | 1.049 | 50% | 5 | 131072 |
| 3 | 0 | | | | | √ | | 0.823 | 85% | 1036 | 131072 |
| 4 | 0 | | | | | √ | | 1.108 | 50% | 1 | 131072 |
| 4 | 3 | | | | √ | | | 0.447 | 85% | 2499 | 131072 |

To ensure each model can run in a reasonable amount of time, only the top 65,536 most abundant motifs in a biopsy are used.

Every motif was scored on the basis of these features using a logistic function that calculates the probability that a motif was derived from a HGSOC-associated repertoire. We used the logistic function because of its widespread use and simplicity, and because it models the outcome of a two-category process. The first step was to compute a biased, weighted sum of the features, referred to as a logit. The logit for the $i$th motif is represented as $l_i$.

$$l_i = \beta_0 + \beta_1 \cdot f_1 + \beta_2 \cdot f_2 + \ldots + \beta_{15} \cdot f_{15} + \beta_{16} \cdot f_q \qquad (1)$$

The bias term $\beta_0$ along with the weights $\beta_1$ through $\beta_{16}$ are the parameters of the model and were fit using gradient descent optimization techniques as described below. Every motif was scored using the same values for the weights and bias term.

Once the logit was computed, the value was passed through the sigmoid function to obtain a probability value between 0 and 1 for the $i^{\text{th}}$ motif.

$$P_i = {}^1\!/_{1 + e^{-l_i}} \tag{2}$$

## Aggregating motif probabilities (multiple instance learning)

To predict whether a repertoire was derived from HGSOC or healthy ovarian tissue, the probabilities assigned to each motif must be aggregated into a single value that predicts the repertoire-level label. This problem of predicting a label for a set of objects from the scores for the individual objects in the set can be formally described as MIL [23]. According to the standard assumption of MIL, at least one motif from HGSOC-derived repertoires must have a high probability, while none of the motifs from healthy tissue-derived repertoires should have a high probability. This assumption can be implemented by simply taking as the repertoire score the maximum score over all motif scores in the repertoire. Thus, the probability that repertoire $j$ is tumor-derived given the individual motif probabilities was computed as:

$$P_{\text{tumor}}^{(j)} = \text{Max}\{P_1, P_2, P_3, \ldots\} \tag{3}$$

Using 0.5 as the threshold score, a repertoire is predicted to be tumor-derived when at least one motif is scored with a probability $\geq 0.5$. The predictions from Eq (3) were used to fit the model's parameters using Cohort I.

## Parameter fitting

Values for the bias term $\beta_0$ and weights $\beta_1$ through $\beta_{16}$ were selected to maximize the probability that each prediction from Eq (3) is correct, assigning tumor-derived repertoires a probability close to 1 and healthy ovary-derived repertoires a probability close to 0, using the following objective function:

$$\ln L = \sum_j y_{\text{tumor}}^{(j)} \cdot \ln P_{\text{tumor}}^{(j)} + (1 - y_{\text{tumor}}^{(j)}) \cdot \ln (1 - P_{\text{tumor}}^{(j)}) \tag{4}$$

where $y_{\text{tumor}}^{(j)}$ Eq (3) introduces a nonlinear operation into the model, resulting in a model that cannot be fitted using standard optimization techniques for logistic regression. Thus, gradient optimization techniques were used, such that each parameter is iteratively adjusted along the gradient in a direction that maximizes the log-likelihood, which in turn maximizes the likelihood that each prediction is correct (Fig 2).

Because gradient optimization techniques are sensitive to the initial values of the bias and weight terms, the initial values must be carefully selected. As is typical, the bias term $\beta_0$ was initialized to 0. The weight values $\beta_1$ through $\beta_{16}$ were initialized using two different distributions, depending on the feature. Weight values $\beta_1$ through $\beta_{15}$ are for Atchley factors, and initial values were sampled from $W \sim \frac{1}{\sqrt{2}} \mathcal{N}(0, \frac{1}{15})$. The weight value for $\beta_{16}$ is for the relative abundance of each motif, and an initial value was sampled from $W \sim \frac{1}{\sqrt{2}} \mathcal{N}(0, 1)$. This initialization scheme ensures that the contribution from the 15 Atchley factors has the same expected magnitude as the contribution from the motifs' relative abundances.

Next, the Adam optimizer, a gradient descent-based optimizer, was run for 2,500 iterations with a step size of 0.01 [32]. Default values for the other Adam optimizer settings were used ($b_1$ = 0.9, $b_2$ 0.999, $\varepsilon = 10^{-8}$). A limitation of gradient descent-based methods is there is no guarantee of finding the globally optimal solution. Although the logistic model is a linear model, the

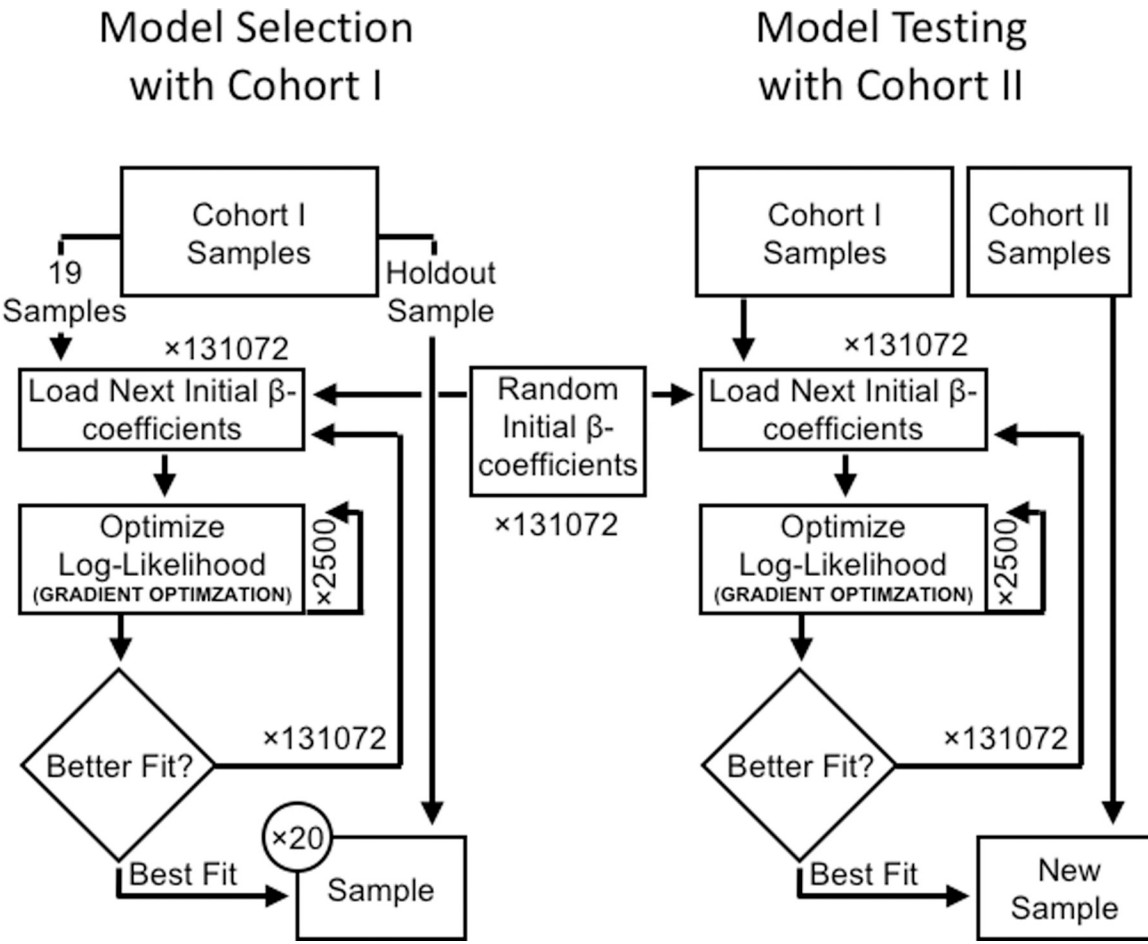

**Fig 2. Workflow for model selection and parameter fitting.** (a) Left Panel: The diagram shows how cohort I is used to train and evaluate each model. Model performance is evaluated by an exhaustive 1-holdout cross-validation using only cohort I. (b) Right Panel: The diagram shows how the best performing model is evaluated with unseen test data. The best performing model is refitted to all the samples in cohort I, and then used to score the test samples from cohort II. The same random initial β-coefficients are reused from (a) when refitting the best performing model in (b).

motif probabilities were aggregated together in a non-linear fashion, and multiple local minima could exist. To address this, $2^{17} = 131,072$ runs of Adam optimization, each starting from different initial values as described in the previous paragraph, were used, and the best fit solution over all runs was used to classify new samples. By identifying the best fit to the training data across a huge number of runs, we were attempting to find the globally optimal solution.

## Overfitting

Overfitting is always a concern with any statistical classifier. We previously found that L1/L2 regularization and dropout worsened the performance of our approach, perhaps because of the highly non-linear characteristics of our model [33]. Thus, we did not apply them in this study, but we did apply early stopping, as described below, which we previously found to significantly improve the model's performance.

## Model selection and testing

We used Cohort I for model selection and validation by patient-hold-out cross-validation (Fig 2, left panel). The same initial values for the weights $\beta_1$ through $\beta_{16}$ were reused with each cross-validation, ensuring the only variation between runs was due to the patient sample being held out, and not because each cross-validation used differential initial values for the weights $\beta_1$ through $\beta_{16}$. We evaluated multiple models (Table 2) and selected as the best model the one with the lowest average negative log-likelihood on patient hold-out cross-validation. Briefly, we considered motifs of either three or four amino acid residues. We also considered gaps, where we allowed either one or two amino acid residues from the CDR3 to be skipped when assembling each possible motif. Additional features and modifications to the model were considered, as indicated in Table 2. For each model, the optimal number of gradient optimization steps was determined by examining the average log-likelihood of each model at each training step in the patient-holdout cross-validation.

To account for model selection bias, the phenomenon whereby we identified a model that performs well on Cohort I without having discovered a generalizable signal, we evaluated the selected model on the test cohort, Cohort II (Fig 2, right panel). The weights and bias term for the best model identified via cross-validation on Cohort I were refit using all 20 Cohort I samples, and then Cohort II samples were scored.

The code is available here: https://github.com/jostmey/MaxSnippetModelOvarian.

## Results

### Cohort I

The best performing model by patient-holdout cross-validation on Cohort I used a motif of three amino acid residues and allowed for a single gap (Table 2). Under that model, the average number of motifs per tumor sample was 7,683.3, and the average number of motifs per healthy sample was 6,154.2. The largest number of motifs in any sample was 13,277. The best average log-likelihood was observed at the last (2,500th) gradient optimization step. The model correctly classified 95% (19/20) of held-out samples with an average log-likelihood of 0.332 bits (Fig 3A). The model correctly classified all healthy ovarian samples, giving a specificity of 100%, although one was quite close to the threshold score of 0.5. The model correctly classified all but one tumor sample, giving a sensitivity of 90%. To estimate the probability of correctly classifying 19 of 20 samples by chance, we performed a permutation analysis with 20 permutation runs. For each permutation, the sample labels were permuted and then patient-holdout cross-validation was performed. Early stopping was applied. The classification accuracies of all 20 permutations were $< 95\%$, allowing us to assign $p < 0.05$ to the observed accuracy (Table 3). The average log-likelihood over all permutations was 0.993 bits, and the average accuracy was 55%.

To discern the features that increase the probability of a HGSOC categorization, we examined the model weights across all 20 cross-validation runs (Fig 3B). The weights reveal how each Atchley factor contributes to the score and the relative importance of each position in the motif. Motifs with a positively charged, hydrophilic residue that tends to participate in alpha-helices in position 1, followed by a small residue that tends to participate in bends and coils in position 2, followed by a large, positively charged residue in position 3 will be scored by the model with a high probability of deriving from a HGSOC-associated repertoire. The weight for the relative abundance of the motif is positive indicating that more abundant motifs would have a higher probability than less abundant motifs.

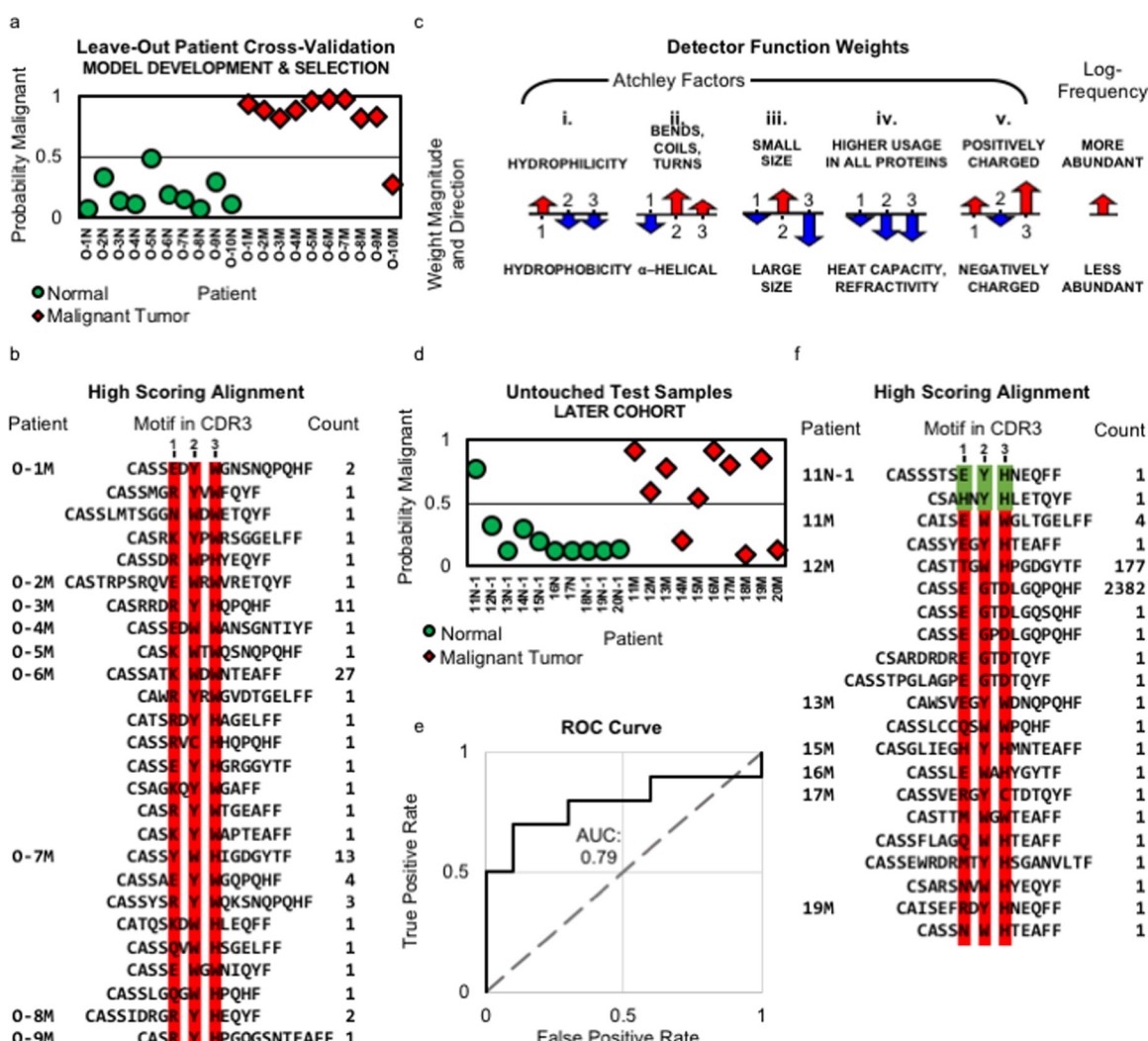

**Fig 3. Results.** (a) Classification results obtained by leave-out cross-validation for each patient in Cohort I. (b) Illustration of the classifier weights averaged across all 20 cross-validation runs (error bars for the standard deviation are omitted because the range was too small to plot relative to the size of each arrow). For each of the five Atchley factors, the weights are shown for the three residue positions. The weight for the log-frequency of the receptor is also shown. Positive weight values are shown pointing up, and negative weight values are shown pointing down. The length of the arrow corresponds to the weight's magnitude. (c) All motifs with a score above 0.5 (middle column) are shown for the 20 patient samples. Each motif is shown in the context of its respective CDR3. The leftmost column indicates the patient and the right most column indicates the number of times the motif is observed in the sample. (d) Classification results obtained on Cohort II test samples. (e) The ROC curve shows true and false positive rates for different thresholds of a positive diagnosis based on the model applied to Cohort II. The area under the curve is 0.79. (f) All motifs with a score above 0.5 (middle column) shown for the 20 patient samples in Cohort II. Each motif is shown in the context of its respective CDR3. The leftmost column indicates the patient and the right most column indicates the number of times the motif is observed in the sample.

We aligned the high scoring motifs from each holdout sample and present them within the context of the CDR3 sequences from which they originated (Fig 3C). The motifs varied in terms of their component residues, but a restricted set of amino acids was observed at each position. Amino acids Glutamic acid, Lysine, and Arginine were common in position 1, Tryptophan and Tyrosine were common in position 2, and Histidine and Tryptophan were common in position 3. We also determined the number of times each CDR3 appeared in each sample and noted that most of them appear only once. None of the CDR3 sequences are shared across patients.

**Table 3. Permutation results.** Each row corresponds to a single permutation of the Cohort I data set, indicated in column 1. The second column shows the loss averaged over all patient-hold-out cross-validations. The third column shows the classification accuracy over all patient-hold-out cross-validations. The fourth column shows the fitting step, out of 2500, at which the lowest average loss was observed.

| Run | Average Loss | Classification Accuracy | Early Stopping Step |
|---|---|---|---|
| 1 | 0.972 | 55% | 132 |
| 2 | 1.07 | 50% | 3 |
| 3 | 1.021 | 50% | 3 |
| 4 | 1.06 | 50% | 1 |
| 5 | 1.055 | 50% | 5 |
| 6 | 1.038 | 60% | 471 |
| 7 | 0.77 | 85% | 503 |
| 8 | 1.03 | 50% | 209 |
| 9 | 0.964 | 65% | 295 |
| 10 | 1.041 | 30% | 245 |
| 11 | 1.011 | 50% | 73 |
| 12 | 1.008 | 55% | 158 |
| 13 | 1.076 | 50% | 4 |
| 14 | 0.63 | 85% | 2497 |
| 15 | 1.012 | 50% | 34 |
| 16 | 1.042 | 50% | 4 |
| 17 | 1.044 | 50% | 4 |
| 18 | 1.043 | 50% | 10 |
| 19 | 1.089 | 50% | 44 |
| 20 | 0.891 | 70% | 1005 |
| **Average** | 0.99335 | 55% | |

## Cohort II

Given the potential for overfitting and model selection bias, we assessed the model's performance on samples not used for model selection or parameter fitting, i.e., on Cohort II. After selecting the best performing model using cross-validation on Cohort I, as described above, we then refit the parameters of the selected model using all 20 Cohort I samples using 2,500 gradient optimization steps, which was determined to be the optimal number of steps in the cross-validation (Table 2). The resulting weights $\beta_1$ through $\beta_{16}$ appear indistinguishable from those in Fig 3C.

The newly fitted model was then applied to Cohort II and correctly classified 80% (16/20) of the samples with an average log-likelihood fit of 0.821 bits. The model correctly classified all but one healthy ovarian sample (specificity 90%) and misclassified three tumor samples (sensitivity 70%) (Fig 3D). The area under the Receiver Operating Characteristic (ROC) curve was 0.79 (Fig 3E).

We aligned the high scoring motifs from the Cohort II samples and present them within the context of the CDR3 sequences from which they originated (Fig 3F). As with the Cohort I motifs, the amino acid residues present at each position vary, but the variability is restricted to a subset. As with the Cohort I samples, amino acids Glutamic Acid and Lysine are common in position 1, Tryptophan and Tyrosine are common in position 2, and Histidine, and Tryptophan are common in position 3. In contrast, Arginine was common in position 1 of Cohort I motifs but is found in position 1 of only one Cohort II motif, and Aspartic Acid is common in position 3 of Cohort II motifs but was not observed in position 3 of Cohort I motifs. As with

Cohort I, we found that the majority of CDR3s containing high-scoring motifs were present only one time in their sample.

## Discussion

We previously hypothesized that T cell responses against antigens shared among cancer patients might enable development of a new approach to cancer detection [25]. Shared tumor antigens are not favored for antigen-targeted immunotherapy where the goal is to elicit such a high degree of tumor-cell killing that the tumor is eradicated. In that case, antigens with expression patterns highly-restricted to the tumor and that are targeted by high-affinity TCRs are needed. For cancer detection, however, it is only necessary that the corresponding TCRs be present in patients with the cancer and not in those without or that they be present with an elevated abundance in those with cancer relative to those without.

To determine whether such T cell responses might enable cancer detection, we first sought to develop a method for identifying the corresponding TCRs that didn't require knowledge of the target antigens and didn't rely on the assumption that T cells responding to a common target would express TCRs with the same amino acid sequence. To accomplish this, we developed the method described here, converting amino acid sequences into numerical vectors whose components correspond to amino acid biophysicochemical values, such as charge, and applying multiple instance learning. In all cases in which the method has been applied, it has identified a motif that can distinguish the tissue or patient groups of interest with solid performance [25, 33]. We hypothesize that TCRs bearing these motifs have overlapping antigen binding profiles and are concentrated in cancer tissue due to the presence of a common antigen there. This is a hypothesis that will have to be tested experimentally, but the strong classification performance of the motifs warrants further study, despite uncertainty regarding any shared antigen specificity.

In our first application of this method to TCRs, we considered motifs of four residues and did not allow gaps [25]. Additionally, we took the natural logarithm of the motif relative abundance term. Taking that same model and fitting the weight values on Cohort I, we obtained a classification accuracy of 90% with a likelihood error of 0.666 (Table 2). To determine whether we could improve the performance, we explored additional models not considered in our prior work (Table 2). The best performing model used a three-residue motif allowing for one gap and achieved a classification accuracy of 95% with a likelihood error of 0.332 (Table 2). Thus, while the approach has produced good results across multiple cancer types, each one has required optimization of the motif representation to obtain the best performance. Additional innovation to the modeling approach is required to produce a method that works across multiple cancer types without this customization.

Whenever multiple models are evaluated on the same data and the best performing model is selected, model selection bias can occur. To determine the extent of model selection bias in our Cohort I result, we evaluated the selected model's performance on Cohort II, which is wholly unseen (i.e., not used for parameter fitting or model selection). The classification accuracy on Cohort II is 80% with a likelihood error of 0.821. Reduced performance on test data is expected, and these results indicate that the model has identified a signal that is expected to generalize to new samples with 80% accuracy.

We have applied the method to three cancer types and in each case identified a distinct biophysicochemical motif. While for breast cancer, all receptors bearing the motif were of high abundance, and in some cases were the top most abundant clone, for colorectal cancer, all but a few of the motif-bearing clones were of low abundance [25]. In the case of ovarian cancer, we again observed that motif-bearing clones are of low abundance, and in fact, in all but a few

cases, the corresponding CDR3 sequences were observed in the sample only a single time. While this is perhaps surprising, we note that frozen tissue was used for the colorectal samples in our prior study, while the ovarian samples in this study were all formalin-fixed paraffin-embedded samples that had been collected between 2009 and 2016. The samples are therefore likely subject to significant DNA damage and to have significantly reduced sequence coverage of target regions [34]. It seems unlikely that the motif identified by our approach is purely an artifact given that it correctly classified 80% of the Cohort II samples. Taking the data at face value, it appears the motifs that mark repertoires as being HGSOC-associated are found in low frequency clones.

While our previous results demonstrate that TCR repertoires from TILs can be distinguished from adjacent healthy tissue repertoires by the presence of TCRs bearing specific, biophysicochemical motifs in their antigen binding regions, our current results go further by demonstrating that TILs repertoires from women with HGSOC can be distinguished from ovarian tissue-associated repertoires from women with healthy ovaries. Thus, in this case, we are distinguishing women with cancer from women without cancer, which is the classification task that is directly relevant to cancer detection. Despite this significant advance over the prior work, however, there are still several limitations that must be addressed. First, the HGSOC samples used in this study were primarily from women with stage III or IV disease. It is critical to determine whether this or another signature can be detected at early stages of disease, particularly before the appearance of invasive disease. Second, to have any potential utility for cancer detection, the signature must be detectable in tissue collected by minimally invasive means. That typically means blood. While the overlap between TILs T cell repertoires and the peripheral T cell repertoire has been shown to be relatively low, it is much higher, with as much as ~50–60% overlap, when the CD8+PD-1+ subset of peripheral T cells is sorted [35–40]. Furthermore, the specific antigens recognized by this subset were similar to that of the TILs population [40]. Thus, it is reasonable to expect that a TCR signature found in the tissue can be detected in this or another peripheral T cell subset.

An additional potential utility of our approach is in the diagnosis of women who present with an ovarian mass. Thus, it will be essential to assess the signature on benign ovarian tumors, as well as on ovarian cancers of other types, to determine whether the signature presented here is present in those cases or whether these have their own unique signature.

Taken together, our current and prior results indicate that TCR-based biomarkers have potential utility for cancer detection. They justify further studies on larger patient cohorts designed to improve the generalizability of the signature with a particular focus on blood samples from patients with early stage disease. Additionally, they justify application of this method in other cancer types, such as pancreatic cancer, where, like ovarian, the need for early detection methods are particularly critical.

## Author Contributions

**Conceptualization:** Lindsay G. Cowell.

**Data curation:** Jared Ostmeyer, Elena Lucas.

**Formal analysis:** Jared Ostmeyer.

**Funding acquisition:** Lindsay G. Cowell.

**Methodology:** Jared Ostmeyer, Scott Christley, Lindsay G. Cowell.

**Software:** Jared Ostmeyer.

**Supervision:** Scott Christley, Lindsay G. Cowell.

**Validation:** Jared Ostmeyer, Elena Lucas.

**Visualization:** Jared Ostmeyer, Lindsay G. Cowell.

**Writing – original draft:** Jared Ostmeyer.

**Writing – review & editing:** Elena Lucas, Scott Christley, Jayanthi Lea, Nancy Monson, Jasmin Tiro, Lindsay G. Cowell.

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
