## [Decision Letter · Decision Letter 0]

16 Jan 2020

PONE-D-19-32179

Biophysicochemical Motifs in T-cell Receptor Sequences as a Potential Biomarker for High-Grade Serous Ovarian Carcinoma

PLOS ONE

Dear Dr. Cowell,

Thank you for submitting your manuscript to PLOS ONE. After careful consideration, we feel that it has merit but does not fully meet PLOS ONE’s publication criteria as it currently stands. Therefore, we invite you to submit a revised version of the manuscript that addresses the points raised during the review process.

We would appreciate receiving your revised manuscript by 30 days. To enhance the reproducibility of your results, we recommend that if applicable you deposit your laboratory protocols in protocols.io, where a protocol can be assigned its own identifier (DOI) such that it can be cited independently in the future. For instructions see: http://journals.plos.org/plosone/s/submission-guidelines#loc-laboratory-protocols

We look forward to receiving your revised manuscript.

Kind regards,

David Wai Chan, Ph.D.

Academic Editor

PLOS ONE

2. To comply with PLOS ONE submissions requirements, please include your ethics statement ('This study was approved by the UTSW IRB, study number STU-2018-0239, with a waiver of consent because anonymized, archived, FFPE tissue was used') in the Methods section of your manuscript.

"This project was supported by funding to LGC from UT Southwestern Medical Center, Be the Difference Foundation, Commercial Real Estate Women of Dallas (CREW Dallas), and an anonymous donor."

"NO authors have competing interests."

We note that you received funding from a commercial source: 'Commercial Real Estate Women of Dallas (CREW Dallas)'

Additional Editor Comments (if provided):

This study is interesting and there are some minor points needed for verification. As the reviewer suggested, it's nice to extend the approach to distinguishing women with or without HGSC by using ovarian tissues from non-cancer patients in their training set.

Reviewers' comments:

Reviewer's Responses to Questions

**Comments to the Author**

1. Is the manuscript technically sound, and do the data support the conclusions?

Reviewer #1: Yes

Reviewer #2: Yes

2. Has the statistical analysis been performed appropriately and rigorously? 

Reviewer #1: I Don't Know

Reviewer #2: Yes

3. Have the authors made all data underlying the findings in their manuscript fully available?

Reviewer #1: No

Reviewer #2: No

4. Is the manuscript presented in an intelligible fashion and written in standard English?

Reviewer #1: Yes

Reviewer #2: Yes

5. Review Comments to the Author

Reviewer #1: This study builds on previous work by this group that T cell receptor repertories (TCR) can be used to identify breast and colon cancer. This study used methods previously established to identify TCR motif in T cells that can distinguish between normal and HGSOC. The authors highlighted an important limitation that the study only examined advanced stage disease. Early stage disease and TILs from blood also need to be examined. The authors should also include in the discussion that performance of the TCR motifs identified in the HGSOC samples should also be compared to those present in benign tumor tissues.

Please provide further information on the TILs sequencing method used. Tissue used from paraffin blocks but is not clear if sequencing is performed at the DNA or amino acid level? is not clear if ovarian and Fallopian tissue was analysed separately or combined in the cases where both were available. Is any information known about the number if TILS present in the tissues.

Minor correction

Please include abbreviation for LSIL in Table 1

Reviewer #2: In their recent manuscript titled “Biophysicochemical Motifs in T-cell Receptor Sequences as a Potential Biomarker for High-Grade Serous Ovarian Carcinoma” Ostmeyer et al. further developed the concept of disease classification based on T-cell receptor (TCR) repertories analysis they put forth in their 2019 Cancer Research paper. This time they attempted to demonstrate specific biophysiochemical motifs in the TCR of tumour infiltrating lymphocytes (TIL) in ovarian tumour exist. Those motifs can help distinguish women with and without high grade serous carcinoma (HGSC). By incorporating healthy women in the training set the motifs identified should be able to be generalized which may help cancer detection in the future.

This is an excellent study probing an interesting possibility of cancer detection. Although the road to application may still be long ahead, the authors managed to demonstrate the feasibility of the principle.

Major concerns

1. Differentiating ovarian cancer tissue form normal ovary is generally not a problem. This study extended their approach to distinguishing women with or without HGSC by using ovarian tissues from non-cancer patients in their training set. This is perhaps the most significant difference from their previous papers and is one step further towards the goal of using TCR repertoire for cancer detection. However, the team still need to demonstrate this technology can help to detect cancer (1) before invasive disease arise; and (2) in samples convenient taken e.g. blood. Until then the technique is no better than current markers such as IHC.

2. The authors focused on HGSC but when a woman is diagnosed with ovarian cancer perhaps it more important to distinguish whether it’s HGSC vs other type I e.g. clear cell which affect management options and prognosis. Currently for equivocal cases IHC markers are used but there are some exceptions. I wonder TCR repertoire can help on this problem.

3. As mentioned in the discussion, TILs repertories and peripheral T cells repertoires are rather different. Although sorting CD8+PD-1+ cells may improve the representation of repertories within the tumour, I expect the accuracy of classification when using peripheral blood will be further reduced. Would increasing sequencing depth help on this? How will be the cost of applying this technology?

4. Can the learning process discover associations between certain motifs with clinical parameters such as chemoresistance, responsiveness towards immunotherapy?

Minor concerns

1. “Additionally, they justify application of this method in other cancer types, such as pancreatic cancer, where the need for early detection methods are particularly critical.” I think ovarian cancer is also a type that early detection is very much desired.

6. PLOS authors have the option to publish the peer review history of their article (what does this mean?). If published, this will include your full peer review and any attached files.

Reviewer #1: No

Reviewer #2: No

---

## [Author Response · Author response to Decision Letter 0]

20 Jan 2020

POINT-BY-POINT RESPONSE

Journal Requirements – the item numbers correspond to those in request email

1. Changes have been made according to the style requirements. These do not show in track changes.

2. The ethics statement has been added to the first paragraph of the Materials and Methods section, in the Datasets subsection (lines 122-124).

3. The data are freely available from the VDJServer Community Data Portal as described in the Materials and Methods section on lines 158-162. The code is available from GitHub per the link now provided on line 331.

5. No change has been made, because CREW Dallas is NOT a commercial entity. It is a 501c3. Please let me know if some change is still required.

6. The ORCID iD has been updated. 

7. The figure tiff files have been converted using PACE. 

Reviewer Comments – in the order they appear in the emailed reviews. Reviewer comments in italics; our response in normal font.

Reviewer #1

The authors should also include in the discussion that performance of the TCR motifs identified in the HGSOC samples should also be compared to those present in benign tumor tissues.

This has been added in the Discussion section, lines 490-493.

Please provide further information on the TILs sequencing method used. Tissue used from paraffin blocks but is not clear if sequencing is performed at the DNA or amino acid level? is not clear if ovarian and Fallopian tissue was analysed separately or combined in the cases where both were available. Is any information known about the number of TILS present in the tissues. 

All of these questions are now answered in the Materials and Methods section, lines 134-138.

Please include the abbreviation for LSIL in Table 1.

Clarification of this abbreviation has been added to the Table 1 footnote, along with the other abbreviations.

Reviewer #2

… the team still need to demonstrate this technology can help to detect cancer (1) before invasive disease arise; and (2) in samples convenient taken e.g. blood. 

We previously mentioned that the use of primarily stage III and IV samples is a limitation of our study, and that “It is critical to determine whether this or another signature can be detected at early stages of disease” (Discussion section, lines 481-482). We have now added the phrase “particularly before the appearance of invasive disease” to emphasize the point made by the reviewer, which is a very important one.

We had previously included a discussion of the need to demonstrate the performance of the technology in blood (Discussion section, lines 482-488). We believe this already addresses the reviewer comment but would be happy to make any suggested changes or additions to this text.

… perhaps it more important to distinguish whether it’s HGSC vs other types

We have added text to the Discussion section (lines 490-493) to clarify that in women who present with a pelvic mass, determining the nature of the mass is an important clinical question and that the utility of our approach for this task needs to be assessed. 

Additional clarification: we focused on HGSOC because it is the most common type of ovarian cancer. We agree with the reviewer that, when a woman is diagnosed with ovarian cancer, it is important to determine what type of cancer it is, or whether it is even cancer (versus a benign tumor). This is currently done by means requiring a biopsy. The ability to make this determination with blood biomarkers would have clinical benefit. However, we are focused on early detection in the context of screening the general population, a clinical application that places different requirements on a molecular biomarker than diagnosis. It is for that reason that we focused on the most common ovarian cancer type and whether its presence could be distinguished from cancer-free women. However, we agree that the reviewer’s suggested application is an important one that we hope to pursue in future studies. 

I expect the accuracy of classification when using peripheral blood will be further reduced. Would increasing the sequencing depth help on this? How will be the cost of applying this technology?

We agree that a straight application of our current method to blood would be expected to have reduced accuracy. We anticipate the need to refit the model parameter that corresponds to motif relative abundance. Additional changes may also be necessary. We are in the process of applying for funding that would allow us to test the model on blood and optimize a blood-specific model. We strongly agree that deep, and perhaps ultra-deep, sequencing will be required, at least in the motif discovery stage. Regarding costs, we don’t know the answer to that. We note, however, that Adaptive Biotechnologies has an FDA-approved assay for detecting measurable residual disease for leukemias that is based on this technology, and the test has received approval to be covered by Medicare. This suggests to us, that, if this approach proves to have clinical utility (which will require many many follow up studies on larger sample sizes, using blood, with more heterogeneous control groups, etc), it will be possible to develop a clinical assay that has been optimized to make it cost-effective. 

Can the learning process discover associations between certain motifs with clinical parameters such as …

We believe the answer is yes, if TCR specificity plays a role in the chemoresistance or response to immunotherapy. We are currently in the process of assessing this in the context of response to immune checkpoint blockade but do not yet have an answer.

I think ovarian cancer is also a type that early detection is very much desired.

This is correct, and we have modified the sentence that originally only mentioned pancreatic cancer to read “Additionally, they justify application of this method in other cancer types, such as pancreatic cancer, where, like ovarian, the need for early detection methods are particularly critical.”

---

## [Decision Letter · Decision Letter 1]

11 Feb 2020

Biophysicochemical Motifs in T-cell Receptor Sequences as a Potential Biomarker for High-Grade Serous Ovarian Carcinoma

PONE-D-19-32179R1

Dear Dr. Cowell,

We are pleased to inform you that your manuscript has been judged scientifically suitable for publication and will be formally accepted for publication once it complies with all outstanding technical requirements.

With kind regards,

David Wai Chan, Ph.D.

Academic Editor

PLOS ONE

Additional Editor Comments (optional):

Reviewers' comments:

Reviewer's Responses to Questions

**Comments to the Author**

1. If the authors have adequately addressed your comments raised in a previous round of review and you feel that this manuscript is now acceptable for publication, you may indicate that here to bypass the “Comments to the Author” section, enter your conflict of interest statement in the “Confidential to Editor” section, and submit your "Accept" recommendation.

Reviewer #1: All comments have been addressed

2. Is the manuscript technically sound, and do the data support the conclusions?

Reviewer #1: Yes

3. Has the statistical analysis been performed appropriately and rigorously? 

Reviewer #1: Yes

4. Have the authors made all data underlying the findings in their manuscript fully available?

Reviewer #1: Yes

5. Is the manuscript presented in an intelligible fashion and written in standard English?

Reviewer #1: Yes

6. Review Comments to the Author

Reviewer #1: The authors have addressed my concerns in the revised manuscript. Revised manuscript is is now suitable for publication.

7. PLOS authors have the option to publish the peer review history of their article (what does this mean?). If published, this will include your full peer review and any attached files.

Reviewer #1: No

---

## [Editor Report · Acceptance letter]

19 Feb 2020

PONE-D-19-32179R1 

Biophysicochemical Motifs in T cell Receptor Sequences as a Potential Biomarker for High-Grade Serous Ovarian Carcinoma 

Dear Dr. Cowell:

I am pleased to inform you that your manuscript has been deemed suitable for publication in PLOS ONE. Congratulations! Your manuscript is now with our production department. 

With kind regards,

on behalf of

Dr. David Wai Chan 

Academic Editor

PLOS ONE